# Release of Microplastics from Reusable Kitchen Plasticware and Generation of Thermal Potential Toxic Degradation Products in the Oven

**Juri Jander** [1]**, Darius Hummel** [1] **, Sophie Stürmer** [1]**, Adrian Monteleone** [1]**, Tizian Neumaier** [1]**, Felix Broghammer** [1]**,
Uta Lewin-Kretzschmar** [2]**, Thomas Brock** [2]**, Martin Knoll** [3] **and Andreas Siegbert Fath** [1,4,*]

[1] Faculty of Medical and Life Sciences, Hochschule Furtwangen, 78054 Villingen-Schwenningen, Germany;
juri.jander@gmail.com (J.J.); d.hummel@uni-hohenheim.de (D.H.);
sophie.katharina.stuermer@hs-furtwangen.de (S.S.); adrian.monteleone@hs-furtwangen.de (A.M.);
tizian.neumaier@hs-furtwangen.de (T.N.); felix.broghammer@hs-furtwangen.de (F.B.)

[2] Employer's Liability Insurance Association of Raw Materials and Chemical Industry—(Competence Center
Hazardous Substances and Biological Agents), 06237 Leuna, Germany;
uta.lewin-kretzschmar@bgrci.de (U.L.-K.); thomas.brock@bgrci.de (T.B.)

[3] Department of Earth and Environmental Systems, University of the South, Sewanee, TN 37383, USA;
mknoll@sewanee.edu

[4] Faculty of Medical and Life Sciences, Institute of Applied Biology, Hochschule Furtwangen,
78120 Villingen-Schwenningen, Germany

[*] Correspondence: fath@hs-furtwangen.de

**Abstract:** Plastics are one of the most important technical materials at present, yet they are associated with a whole series of environmental problems such as micro-and nanoplastics or their plasticizers, which have become increasingly relevant in recent years. While there are many studies that focus on microplastics (MPs) introduced into the human body through commercially produced food, there are nearly none that consider the MPs we ingest through homemade food made with plastic kitchen utensils such as mixing bowls. To investigate this, samples were obtained by exposing different plastic bowls made of acrylonitrile–butadiene–styrene (ABS), polypropylene (PP), melamine, polyethylene (PE), polystyrene (PS), and styrene–acrylonitrile (SAN), to mechanical stress and then analyzed via infrared spectroscopy. This not only raises the question of whether microplastics are incorporated into foods but also the extent to which the degradation products produced by thermal stress in an oven could play a toxicological role. Degradation products were generated by pyrolysis and analyzed afterwards using gas chromatography mass spectrometry. There were differences in the number of microplastic particles abraded by the different types of plastic, with the most consisting of melamine (898 particles) and the least consisting of low-density polyethylene (331 particles). There were also differences in the number and relevance of the thermal degradation products for the different plastics, so that a human toxicological assessment would have to be evaluated in further work.

**Keywords:** microplastic; microplastic analysis; kitchen plastic bowl; pyrolysis; GC-MS; abrasion

## 1. Introduction

Since the industrialization of plastics in the 1950s, their applications have also increased, and they have become an important technical material in the world, but the waste they produce has become an increasing global problem over the years. The high durability of plastic makes it highly resistant to degradation; hence, disposing of plastic poses a big challenge [1]. Therefore, plastics accumulate in the environment and landfills, and it is projected that by the year 2050, 12 billion metric tons of plastics will be distributed throughout both settings [2]. Plastics that end up in the environment are broken down by mechanical [3], chemical [4], or biological [5] processes into smaller pieces. For that reason, microplastic can be passed into the human body through the consumption of contaminated

food. It has been suggested that exposure to microplastic through the consumption of food can influence growth, behavior, and histopathological changes [6]. It is unknown if the ingestion of microplastics released from take-out food containers poses a risk to human health [7], but recently published studies on mice [8] demonstrate the ability of polystyrene microplastic particles to cross the blood–brain barrier and affect the immune system. Plastics even present a risk for human health as sorbents for different drugs such as X-ray contrast agents [9], antibiotics [10], or trace elements [11,12]. While there are many studies that focus on MPs introduced into the human body through commercially produced food [13–16], there are nearly none that consider the MPs we ingest through homemade food made with plastic kitchen utensils such as mixing bowls. This not only raises the question of whether microplastics are incorporated into foods but also the extent to which the degradation products produced by thermal stress in an oven could play a toxicological role.

To investigate these questions, six commercially available plastic bowls and a glass bowl were tested for abrasion with a mixing machine. In addition, the effect on the abrasion of plastic and microplastic particle count of granular or crystalline substances such as sugar or salt were investigated. Fourier transform infrared spectrometer (FTIR-spectrometer) analysis was used to count and specifically determine the type of plastic particles produced (microplastics over 25 μm). Plastic samples were cut out from the bowls and prepared with a cryo-mill for the determination of volatile degradation products originating between 200 and 250 °C. These products were treated using pyrolysis, analyzed by gas chromatography mass spectrometry (GCMS), and subsequently measured.

## 2. Materials and Methods

### 2.1. Sample Material

For the experiments, we used commercially available kitchen mixing bowls made of six different types of plastic, which were to provide a range of abraded microplastics as well as thermally released pollutants during general baking. The indicated plastic type for each bowl was double checked via IR spectroscopy using the Bruker polymer database. The six bowls were made of the following materials: bowl 1 of styrene–acrylonitrile copolymers (SAN), bowl 2 of melamine, bowl 3 polystyrene (PS), bowl 4 of acrylonitrile butadiene styrene (ABS), bowl 5 of low-density polyethylene (LDPE), and lastly, bowl 6 of polypropylene (PP). The selection of plastics also demonstrates different properties such as the Rockwell hardness of engineering materials, which can play a role in mechanical abrasion (Table 1).

**Table 1.** The range of Rockwell hardness of selected polymers.

| Polymeric Material | Minimum Rockwell Hardness | Maximum Rockwell Hardness |
|---|---|---|
| Acrylonitrile–butadiene styrene (ABS) | 2.00 | 70.00 |
| Low-density polyethylene (LDPE) | 1.00 | 20.00 |
| Melamine | 67.00 | 86.00 |
| Polypropylene (PP) | 1.00 | 20.00 |
| Polystyrene (PS) | 60.00 | 75.00 |
| Styrene–acrylonitrile copolymers (SAN) | 10.00 | 80.00 |

### 2.2. Sample Preparation

To assess the extent to which microplastics can be abraded during everyday uses of kitchen bowls, several steps were taken. First, all six individual bowls were treated in the same manner with a hand mixer (Q4835DE, Bosch, Stuttgart, Germany). The mixer was used for a total of 2 min at 200 rpm, which was intended to abrade microplastic particles. Then, 100 mL of purified water was added to each bowl, which was followed by three repetitions of abrasion runs. An additional sample run was conducted with salt in the bowl in order to gauge the influence of granular ingredients on abrasion. This

additional test was carried out with the ABS bowl (ABS I), in which 5 g of rock salt (Steinsalz, Herbaria, München, Germany) was added to the purified water. This rock salt, unlike marine salt, is completely free of microplastics [12,13]. Furthermore, a glass bowl was included in the analysis as a negative control, whereby the environmental parameters such as contamination from air and the water used were established to normalize the results. To investigate if the detected MP particles were in fact from the plastic bowls, two different blank samples were made. The first was to determine if the particles originated from the hand mixer components. For this, 100 mL of water was mixed for 2 min at 200 rpm in a glass bowl. The second blank sample was made to check if there was any MP contamination in the tap water used. For that, 100 mL of tap water was utilized. After these steps, the water was removed from the bowls, filtered, and the filtrate was prepared for analysis with the Perkin Elmer Frontier Fourier transform infrared spectrometer (Perkin Elmer Frontier and Perkin Elmer Spotlight, FTIR-spectrometer). The sample suspensions were filtered through a high-purity aluminum oxide membrane with a pore size of 0.2 μm (Anodisc™ 25, Diameter 25 mm, GE Healthcare UK limited, Little Chalfont Buckinghamshire, Amersham Place, UK) and was afterwards dried at 40 °C overnight. Then, the particles on the aluminum oxide membrane were ready for further analysis with the FTIR spectrometer (Section 2.3 Microplastic Analysis).

Finally, the kitchen bowl plastics had to be prepared as controls for the thermal decomposition products with the pyrolysis, which was followed by GC-MS. To be able to identify decomposition products that can arise during pyrolysis in the oven, a 2–3 g sample of each of the six-plastic bowls was separated with a heavy metal knife and then processed into powder using a cryogenic mill. To crush different plastic types, 3 sequences of the cryogenic mill were performed: pre-cooling, crushing, and intercooling. Pre-cooling lasted 90 s at a frequency of 5 Hz with one cycle, the crushing lasted 300 s at 30 Hz for two cycles, and intercooling lasted 30 s at 5 Hz for one cycle. To generate the required microplastic particles, 2–3 g of each plastic were comminuted by a one-cycle process using a cryogenic swinging mill (CryoMill, Retsch GmbH, Haan, Germany). In contrast to the other plastics, the parameters described above for the plastic LDPE had to be extended by an additional cycle to produce a finer powder. The powder thus obtained from the various plastic bowls could now be used for analysis with the pyrolysis, which was followed by GC-MS (Section 2.4 Analysis of thermal decomposition products).

*2.3. Microplastic Analysis (FT-IR Spectroscopy)*

In order to determine the specific plastic of the particles on the aluminum membrane, the FTIR spectrometer was used. Sixteen scans per particle were performed to detect individual particles to reduce the noise of the spectral data obtained. Only particles with a diameter larger than 25 μm were included in the analysis, since no qualitative high-quality spectra can be obtained below this limit for this purpose. Due to the technical equipment of the FTIR used, an automated measurement of a complete sample filter with a filter area of 490 mm$^2$ was not possible, as this measurement would be very time consuming and additionally susceptible to measurement interruptions. For this reason, statistical extrapolation from square grid fields to the complete filter had to be applied. For this purpose, three square fields (side lengths of 2 mm) with an area of 4 mm$^2$ were measured for each sample at the same positions of the complete round sample filter (490 mm$^2$), and the plastics were identified and counted. Absorption spectra of the measured particles were created through the microscope, and these were compared with the existing plastic databases using a "search function". Based on the number of particles in these three square areas of 4 mm$^2$ each, an extrapolation was made to the total area of the sample filter 490 mm$^2$ [17]. The calculated and extrapolated particle counts with the error bars are shown in Figure 1.

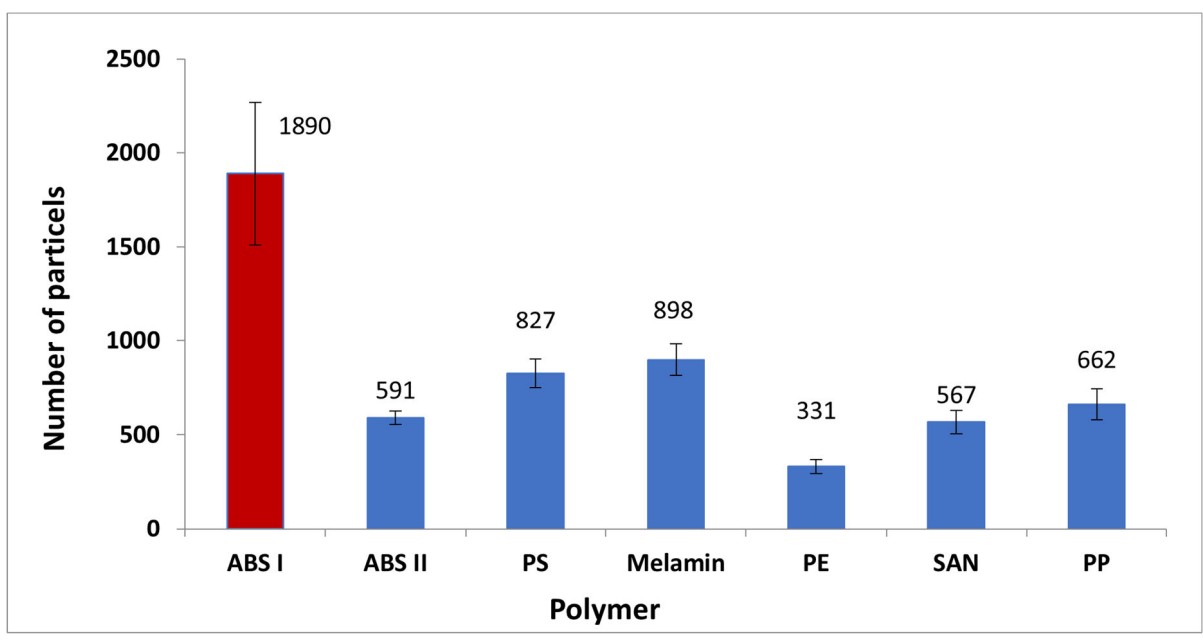

**Figure 1.** FTIR spectrometer microplastic analysis of the different plastic bowls listed according to the type of material. Red bar = use of water and salt. Blue bar = use of water only.

### 2.4. Analysis of Thermal Decomposition Products

For the analysis of the thermal decomposition products resulting from the mixing-derived microplastics during baking, a pyrolysis of polymers was performed, and the products were analyzed afterwards by GC-MS. To detect volatile compounds, 0.5 g of the material samples were weighed, placed into vials, and heated to 200 °C and 250 °C in the thermal block (Liebisch Labortechnik, Bielefeld, Germany). After 30 min, 500 μL samples were taken from the vapor space above the samples with a gas-tight syringe, and then, these were chromatographed and analyzed by mass spectrometry. Under these conditions, water was released from all materials, which is not explicitly mentioned below. For this purpose, a gas chromatograph (6890N, Agilent Technologies, Palo Alto, CA, USA) with split/splitless injector and mass-selective detector (5973N, Agilent Technologies, Palo Alto, CA, USA) was used. The separation was performed with fused silica capillaries (Zebron ZB-624, from Phenomenex) with a length of 60 m, inner diameter of 0.32 mm, film thickness of 1.8 μm, and helium as a carrier gas (0.9 mL/min in constant flow). The heating process was carried out in 4 steps: (1) initial temperature 50 °C (isothermal for 8 min); (2) heating rate I: 5 °C/min to 100 °C (isothermal for 5 min); (3) heating rate II: 10 °C/min to final temperature; and (4) final temperature at 180 °C for 5 min isothermal. The mass spectrometry used electron impact ionization at 70 eV, and the temperatures were 230 °C for the ion source and transfer line.

### 3. Results and Discussion

#### 3.1. Microplastic Analysis (FTIR Spectroscopy)

The results of counting the microplastic particles via FTIR spectrometry of the six different plastic bowls, as well as the bowl made of ABS (ABSI) with the salt treatment, are shown in Figure 1. The IR microscope recognized some particles within the first blank sample, but none of them matched with the references found in the database, which means they were not MP. These particles were probably lime from the faucet. In the second blank sample, only one particle was found, which matched the database. The one particle had a 60% correlation to polyurethane. Due to only one plastic particle in the reference samples for the glass bowl and the used water, which does not correspond to the plastic types used, no normalization of the values in Figure 1 was performed. For the sake of comparison, the samples treated with only water show abraded particles in descending order from

Melamine with 898 particles > PS with 827 > PP with 662 > ABS 591 > SAN with 567 > PE with only 331 particles. The effect of the salt test on the plastic ABS shows an increase from 591 particles (ABS II) of about 320% to 1890 particles (ABS I).

### 3.2. Analysis of Thermal Decomposition Products (Pyrolysis and GC-MS)

The Py-GCMS showed different results for the included plastics, which are described in the next paragraphs. In performing pyrolysis, the heating of the polymers revealed which substances were produced in abundance and in trace amounts as a result of the process.

Acrylonitrile–butadiene–styrene copolymer (ABS). The same spectrum of pollutants was obtained at both temperatures, although the intensities of most pollutants were higher at 250 °C than at 200 °C. The complete table of degradation products for the temperature 200 °C (Table S1) and 250 °C (Table S2) can be found in the Supporting Information. Styrene was the main component detected in the vapor space under these conditions. Furthermore, benzaldehyde, acrolein, n-alkanals (acetaldehyde to n-nonanal), acetophenone, 2-phenylisopropanol, methylstyrene, ethylbenzene, toluene, cumene, 2-methyl-1-propene, and a mixture of other unsaturated aliphatic hydrocarbons (highest single concentration: 1-tridecene) were detected. In low concentrations, there was evidence of acrylonitrile, 4-vinylcyclohexene, and acetone, as well as traces of benzene, ethanol, furan, tert-butanol, 2-butanone, propanenitrile, phenol, and other alkanals and alkyl aromatics.

Low-density polyethylene (LDPE). Under these conditions (Tables S3 and S4), acetaldehyde was detected as the main component in the vapor space, and at 200 °C, acetic acid and formic acid were also detected. Furthermore, n-butane, 2-butanone, and n-alkanals (n-propanal to n-nonanal) were also detected. In low concentrations, there were indications of ethanol, acrolein, acetone, propane, further n-alkanes (n-pentane to n-dodecane), as well as traces of methanol, n-propanol, n-butanol 2-and 3-alkanones and, at 200 °C, dihydro-2(3H)-furanone (butyrolactone).

Polypropylene (PP). The main components detected under these conditions in the vapor space were acetone and, at 200 °C (Table S5), acetic acid and formic acid. The latter could not be detected at 250 °C (Table S6) and acetic acid was detected only in low concentrations. With the exception of some ketones, the concentrations of the other contaminants were higher at 250 °C. A complex mixture of saturated and unsaturated aldehydes (highest concentrations: methacrolein, acetaldehyde), saturated and unsaturated hydrocarbons (highest concentrations: 2,4-dimethyl-1-heptene, 2-methyl-1-pentene, 2-methyl-1-propene, n-pentane), saturated and unsaturated ketones (highest concentrations: 2,4-pentanedione, 2-pentanone, 4-methyl-2-heptanone, methyl isobutyl ketone), and alcohols (highest concentrations: methanol and higher branched alcohols that could not be clearly identified) were detected.

Styrene–acrylonitrile copolymer (SAN). At 200 °C (Table S7), the main components detected in the vapor space under these conditions were styrene and ethylbenzene. Hydrogen cyanide, acrylonitrile, benzaldehyde, acetophenone, phenol, and other aromatic hydrocarbons (e.g., xylenes, cumene, propylbenzene) were detectable in traces, and there was evidence of a complex mixture of unsaturated aliphatic hydrocarbons (C number range approximately C11 to C12). At 250 °C (Table S8), the main components detected in the vapor space were styrene, benzaldehyde, ethylbenzene, and acetophenone. Hydrogen cyanide and phenol were detectable in low concentrations, and there was evidence of 2-phenylpropenal and a complex mixture of unsaturated aliphatic hydrocarbons (C number range approximately C11 to C12). Other aromatic hydrocarbons (e.g., benzene, xylenes, cumene, propylbenzene, methylstyrene), acrylonitrile, acetaldehyde, acetone, and benzonitrile were identified in trace amounts.

Melamine. At both temperatures (Tables S9 and S10), formaldehyde and methanol were detected as the main components in the vapor space under these conditions. Acetaldehyde, methyl formate, and 2,3-butanedione were identified in low concentrations and, additionally, at 250 °C, furfural, furfuryl alcohol, acetic acid, hydroxyacetone, butyrolactone, 2(5H)-furanone, 5-methylfurfural, acetone, and benzaldehyde (which could only

be detected in trace amounts at 200 °C). At 250 °C, there was also evidence of further furan compounds, ketones, and formaldehyde derivatives, which could not be clearly identified.

Polystyrene (PS). A similar spectrum of pollutants was obtained at both temperatures (Tables S11 and S12), although the concentrations of most pollutants were higher at 250 °C than at 200 °C. The main components determined in the vapor space under these conditions were styrene and benzaldehyde. Furthermore, acetophenone, 2-phenylpropenal, ethylbenzene, methylstyrene, cumene, propylbenzene, phenol, phenylacetaldehyde and phenyloxirane were detected. Benzene, formaldehyde, acetic acid, acetaldehyde, benzyl alcohol and a propyl toluene were identified in low concentrations at 250 °C, which could only be detected in the trace range at 200 °C or not at all in the case of acetic acid. In the trace range, there were also indications of n-alkanals, 2-alkanones, benzyl methyl ketone, unsaturated ketones, other aromatic hydrocarbons, and alcohols, in particular at 250 °C.

## 4. Discussion/Conclusions

### 4.1. Microplastic Analysis (FTIR Spectroscopy)

When assessing the results generated, several parameters must be associated with each other and, if possible, correlated. One is the type and hardness of the plastics as well as the influence of friction-enhancing granular or crystalline substances and the resulting effect on the number of microplastics. The resulting number of microplastics influences the potential amount of degradation products generated by thermal stress in the oven. Meaning for real application that abraded mass of plastic particles from the production of for example cake or bread in plastic kitchenware are approximately proportional to their thermal decomposition products in the oven. The amount of the produced microplastic particles is shown for each polymer (Figure 1). This makes it possible to determine which material was the most resistant to mechanical force. The plastics melamine and PS show the highest abrasion without salt treatment, which can be explained by considering the Rockwell hardness, which, compared to the other plastics, is in the high range for melamine (898 particles) between 67.00 and 86.00 and PS (827 particles) at 60.00–75.00 (Table 1). The plastics ABS and SAN show a relatively similar range of Rockwell hardness of 20.00–70.00 (ABS) and 10.00–80.00 for SAN, which is reflected in the only 4% difference in microplastic count. LDPE has the lowest microplastic count with only 331 particles, which can be explained by a very low Rockwell hardness value of 1.00–20.00. Furthermore, the influence of friction-increasing substances such as salt or other components, which could occur in the added ingredients when mixing in a plastic bowl, was tested. It was found that adding salt to the plastic ABS when stirring in water can cause a tripling of the microplastic count from 591 particles to 1890 particles. This means that the type of ingredients and the type of processing in a plastic bowl can also have an influence on the microplastic content.

### 4.2. Analysis of Thermal Decomposition Products (Pyrolysis and GC-MS)

In performing pyrolysis, the heating of the polymers revealed which substances were produced in abundance and in traces. From these results, initial predictions can then be made about the risk of such plastic particles and their degradation products in various baked foods. However, not all parameters needed for a specific toxicological assessment could be considered in this study, and further studies are needed for a complete estimation.

In the case of the ABS polymer, the main substances detected were styrene, benzaldehyde, and acrolein. This is concerning, as all these substances are considered carcinogenic [18,19]. Furthermore, substances such as acetophenone, methylstyrene, and ethylbenzene were found. These substances are also harmful to health, as they can irritate the respiratory system [18,19]. In addition, it was possible to detect traces of substances such as ketones, phenols, alkenes, and aromatics. However, they can be neglected due to their low concentrations. Analysis of the PS polymer revealed mainly benzaldehydes and styrenes as products. Benzaldehydes are only considered harmful to health in higher quantities. In the case of styrenes, there is a risk of them being converted into styrene oxides when ingested [20,21]. These are extremely dangerous, as they are potentially muta-

genic and carcinogenic [18–20]. Substances such as benzene or benzyl alcohol were found in traces.

The results of the analysis of the melamine polymer are extremely worrying, as the major products were the very toxic and carcinogenic substance formaldehyde [18,19] and the toxic and highly hazardous substance methanol [18,19]. This was observed when pyrolysis was carried out at both temperatures of 200 °C and 250 °C. In addition, the presence of acetaldehydes, methyl formates, and 2,3-butanedione could be determined at a temperature of 250 °C. At the temperature of 200 °C, these could only be found in traces. In Germany, it is forbidden to heat products containing melamine to over 70 °C, since formaldehyde is released at this temperature [22]. Regarding the LDPE polymer, the formation of the substance acetaldehyde, which is harmful to health and carcinogenic, was one the main products. Acetic acid was also synthesized in higher quantities. The acid has an irritating effect on mucous tissues [18,19]. Formic acid was also released, which is considered dangerous, especially in high concentrations, as it is irritating to the eyes and can cause respiratory distress if inhaled [18,19]. Butane was also formed, which has a fatiguing effect [23], as well as aldehydes, which can potentially cause allergies [18,19]. Traces of acroleine, propane, ketones, and short-chain alcohols were detected. However, in the second pyrolysis experiment at a temperature of 250 °C, both acetic acid and formic acid were not formed.

Pyrolysis products of SAN polymer have been identified primarily as styrene and ethybenzene. These are both considered hazardous to health, as they can cause cancer. However, when pyrolysis was carried out at 250 °C, acetophenones and benzaldehydes were also formed in larger concentrations. The trace substances formed at both 200 °C and 250 °C included hydrocyanic acid and phenols as well as other aromatic hydrocarbons and acetophenone. More trace substances were found during pyrolysis at 250 °C. When examining the resulting products concerning the polymer PP, acetone as well as formic and acetic acid could be identified in large quantities. Acetone is extremely harmful because it is particularly irritating to the eyes. The same applies to acetic acid. Formic acid is very damaging if it is ingested and, in addition, it has a severe corrosive effect on tissues. In addition, methanol was detected. This is hazardous to health when inhaled, touched, or ingested.

By comparing particle amounts and hazard classifications of main components, we deduce that LDPE seems to be the safest plastic to be used in a common household mixing bowl. Apart from acetaldehyde, which is damaging to organs in higher concentrations, its other components are not as hazardous in comparison. Additionally, the number of particles measured following the abrasion process was the lowest of all available plastics. This study shows the extent to which the various plastics (microplastics) are decomposed by thermal stress to harmful or questionable substances. It is not evident whether these substances volatilize in the oven or accumulate in foodstuffs through adsorption or absorption. As already mentioned, no conclusive toxicological assessment can be given yet, since for this the dose, the exposure times of the intake and the bioaccumulation in the organism of the resulting substances would have to be considered. Thus, studies on the ad- or absorbed or bound substances in the prepared food would have to be carried out, which should be addressed in future research work.

**Supplementary Materials:** The following supporting information can be downloaded at: https://www.mdpi.com/article/10.3390/app12052535/s1, Tables S1–S12: The complete table of degradation products for both temperatures (PDF).

**Author Contributions:** Formal analysis, U.L.-K., T.B.; investigation, J.J., D.H., T.N., F.B.; writing—review and editing, S.S., M.K.; supervision, A.M.; project administration, A.S.F. All authors have read and agreed to the published version of the manuscript.

**Funding:** The article processing charge was funded by the Baden-Württemberg Ministry of Science, Research and Culture and the Furtwangen University in the funding programme Open Access Publishing.

**Acknowledgments:** We would like to thank the summer semester 2021 course of study "Biology & Process technology" in the faculty of medical & life sciences within the lecture "scientific writing" for their contribution to the manuscript.

**Conflicts of Interest:** The authors declare no conflict of interest.

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
