# Peer review of "Release of Microplastics from Reusable Kitchen Plasticware and Generation of Thermal Potential Toxic Degradation Products in the Oven"

_applsci, doi:10.3390/app12052535_

Round 1

Reviewer 1 Report

This is an interesting study. I think microplastics in food utensils are indeed a topic worthy of research. I think this work can be accepted for publication after major revision.

  1. The author needs to improve the resolution of the picture.
  2. I didn't particularly understand how the authors estimated the amount of microplastics using FTIR.
  3. Why would the author include some results in supporting material when there is only one figure in the MS? Also, the support material is not uploaded to the system with MS.
  4. Thermal decomposition is not linked to the calculation of microplastic particles. I am not sure what the significance of studying thermal decomposition is.
